# The effect of the choice of screening test when measuring the prevalence of gambling disorder: A cross-sectional study in Japan

Tatsuya Noda[1☉*], Moritoshi Kido[2☉], Chieko Ito[1,3☉], Toshiyuki Ojima[3]

1 Department of Public Health, Health Management and Policy, Nara Medical University, Kashihara, Nara, Japan, 2 Faculty of Human Sciences, Department of Physical and Mental Health Science, AICHI MIZUHO College, Nagoya, Aichi, Japan, 3 Department of Community Health and Preventive Medicine, Hamamatsu University School of Medicine, Hamamatsu, Shizuoka, Japan

☉ These authors contributed equally to this work.
* noda@naramed-u.ac.jp

## Abstract

This study examines the influence of the selection of screening tests and cut-off scores on the prevalence of gambling disorders by simultaneously administering several tests to the same sample. The survey was conducted online in 2021, with 2,000 respondents distributed equally across two prefectures in Japan. Four screening tests were administered simultaneously: the South Oaks Gambling Screen (SOGS), Problem Gambling Severity Index (PGSI), Lie/Bet questionnaire, and Diagnostic and Statistical Manual of Mental Disorders-5 (DSM-5). The prevalence at the original cut-off scores was markedly different, with the SOGS (10.3%) showing the highest prevalence and the DSM-5 (3.8%) showing the lowest prevalence. Adjusting the cut-off score from 5 to 4 for the SOGS increased prevalence by 2.9%, while changing the PGSI cut-off score from 8 to 7 only increased it by 0.5%. This is the first study in Japan to simultaneously compare the scores for multiple screening tests and cut-off scores regarding gambling disorders. The SOGS screens more individuals with a possible gambling disorder than other measures, and altering the cut-off score significantly affected its prevalence. Selecting appropriate screening tests and cut-off scores is crucial to accurately assessing the prevalence of possible gambling disorders.

## Introduction

Gambling disorder is a persistent problematic behavior that leads to clinically significant impairment and/or distress [1]. Measuring its prevalence in the general population is critical because gambling disorder causes financial problems, damages relationships and health, leads to psychological distress, and lowers work performance and academic achievement [2]. Gambling disorder first appeared in the third edition of the Diagnostic and Statistical Manual of Mental Disorders (DSM-3) published by the American Psychiatric Association in 1980, using the diagnostic name 'pathological gambling' [3]. The DSM-5 provides diagnostic criteria for gambling disorder in the non-substance-related disorder group of substance-related and addictive disorders [1]. The prevalence of problem gambling in the past 12 months varies widely depending on the screening test used, ranging from 0.12% to 5.8% [4]. In the United

**Data availability statement:** The datasets acquired and analyzed in this study are available upon reasonable request. All data files are owned by the KAKENHI Research Team on Addiction under the approval of the Ethics Committee of Nara Medical University. Data requests should be sent to the Nara Medical University Research Staff (g_phdata@naramed-u.ac.jp).

**Funding:** This work was supported by JSPS KAKENHI Grant Number JP17K09092 and JP20K14237. The funder had no role in study design, data collection and analysis, decision to publish, or preparation of the manuscript.

**Competing interests:** The authors have declared that no competing interests exist.

States and Australia, the prevalence of problem gambling began increasing in the late 1980s and early 1990s and peaked in the late 1990s and early 2000s. This was related to the opening of casinos and the introduction and expansion of electronic gambling machines in these countries. However, the prevalence has been declining since the late 1990s in the United States and the early 2000s in Australia [5]. Conversely, the prevalence of problem gambling increased in certain countries and regions during the early to late 2000s. Changes in the gambling environment, including the expansion of internet gambling, are thought to be contributing factors to this increase [4]. In Japan, results from the SOGS-R indicate a prevalence of problem gambling of 8.04% [6]. Nevertheless, no systematic fact-finding surveys have been conducted in Japan to determine whether the prevalence of possible gambling disorder has increased or decreased.

Two screening tests regarding gambling disorders are commonly used worldwide [7]. The first of these is the South Oaks Gambling Screen (SOGS) [8–10]. The SOGS is a self-administered tool that evaluates gambling activity based on frequency, amount spent, problem occurrence, self-assessment, and sense of control. It allows for convenient scoring and assessment across a broad range of problem areas [9,11]. The second tool is the Problem Gambling Severity Index (PGSI) [12,13]. The PGSI evaluates the severity of gambling problems, including frequency of participation, financial impact, loss of control, and effects on daily life, and has high reliability and validity [12]. In Japan, surveys using both the SOGS and the PGSI have been conducted nationally [10,14–19]. While the SOGS was frequently used in the recent past, the use of the PGSI, either alone or together with the SOGS, has recently been increasing [5]

However, prevalence varies widely from survey to survey [5], which occurs well beyond the threshold of the expected variation and despite the surveys using the same sampling methods. These variations in prevalence can be attributed to the (i) selection of screening tests, (ii) setting of cut-off scores, and (iii) survey mode (e.g., face-to-face, mail) [5]. Although these characteristics should be considered when conducting surveys, previous studies on this topic remain scarce. Indeed, although it would be preferable to conduct multiple screening tests on the same sample simultaneously to identify the causes of these fluctuations in prevalence, few such studies have been conducted [20–23]. Hence, research should focus on the differences in screening tests and the interchangeability of cut-off scores [5]. Given this gap in the literature, the main purpose of our study is to investigate the impact of the choice of screening method and cut-off score on the prevalence of possible gambling disorders.

## Materials and methods

We conducted an online survey in two Japanese prefectures with comparable population sizes: Osaka, where an integrated resort including casinos is scheduled to open in 2029 [24], and Fukuoka, where no such resort is planned.

The survey was completed by 1,000 respondents in each prefecture in April 2021. The survey started on 14 April 2021 and ended on 19 April 2021. The 2,000 respondents were registered monitors of ASMARQ Co., Ltd., and the number of respondents was matched to the actual sex and age distribution in each prefecture [25]. The survey collected information on respondents' basic attributes (e.g., age and sex; Table 1) and their experience of participating in eight gambling activities (Table 2) and contained four screening tests. These four tests were the SOGS [8–10], the PGSI [12,13], the Lie/Bet questionnaire (LieBet) [26,27], and the DSM-5 [1,28]. The SOGS, PGSI, and DSM-5 are described above. LieBet, an early screening tool, consists of two questions focusing on lying and chasing behaviors related to gambling. While it can be administered quickly, it must be used alongside other tools for a comprehensive assessment [27]. The time frames for each screening test were adopted from their original

**Table 1. Basic attributes of the respondents.**

| | | Total (male %) | Osaka (male %) | Fukuoka (male %) |
|---|---|---|---|---|
| **Age** | 20–29 | 350 (49.7%) | 179 (49.7%) | 171 (49.7%) |
| | 30–39 | 368 (49.2%) | 184 (49.5%) | 184 (48.9%) |
| | 40–49 | 473 (49.0%) | 239 (49.4%) | 234 (48.7%) |
| | 50–59 | 423 (48.7%) | 222 (49.5%) | 201 (47.8%) |
| | 60–69 | 386 (47.9%) | 176 (48.3%) | 210 (47.6%) |
| | Total | 2000 (48.9%) | 1000 (49.3%) | 1000 (48.5%) |

**Table 2. Number and percentage of people engaged in the eight gambling activities.**

| | | Total participation | Participation within the past year | Participation over a year ago |
|---|---|---|---|---|
| **Lottery** | Total | 1169 (58.5%) | 580 (29.0%) | 589 (29.5%) |
| | Osaka | 580 (58.0%) | 291 (29.1%) | 289 (28.9%) |
| | Fukuoka | 589 (58.9%) | 289 (28.9%) | 300 (30.0%) |
| **Pachinko** | Total | 844 (42.2%) | 246 (12.3%) | 598 (29.9%) |
| | Osaka | 438 (43.8%) | 133 (13.3%) | 305 (30.5%) |
| | Fukuoka | 406 (40.6%) | 113 (11.3%) | 293 (29.3%) |
| **Horse racing** | Total | 674 (33.7%) | 294 (14.7%) | 380 (19.0%) |
| | Osaka | 379 (37.9%) | 173 (17.3%) | 206 (20.6%) |
| | Fukuoka | 295 (29.5%) | 121 (12.1%) | 174 (17.4%) |
| **Sports promotion lottery** | Total | 436 (21.8%) | 234 (11.7%) | 202 (10.1%) |
| | Osaka | 225 (22.5%) | 125 (12.5%) | 100 (10.0%) |
| | Fukuoka | 211 (21.1%) | 109 (10.9%) | 102 (10.2%) |
| **Boat racing** | Total | 299 (15.0%) | 110 (5.5%) | 189 (9.5%) |
| | Osaka | 131 (13.1%) | 58 (5.8%) | 73 (7.3%) |
| | Fukuoka | 168 (16.8%) | 52 (5.2%) | 116 (11.6%) |
| **Casino outside Japan** | Total | 244 (12.2%) | 43 (2.2%) | 201 (10.1%) |
| | Osaka | 134 (13.4%) | 24 (2.4%) | 110 (11.0%) |
| | Fukuoka | 110 (11.0%) | 19 (1.9%) | 91 (9.1%) |
| **Bicycle racing** | Total | 215 (10.8%) | 96 (4.8%) | 119 (6.0%) |
| | Osaka | 100 (10.0%) | 51 (5.1%) | 49 (4.9%) |
| | Fukuoka | 115 (11.5%) | 45 (4.5%) | 70 (7.0%) |
| **Motorcycle racing** | Total | 169 (8.5%) | 76 (3.8%) | 93 (4.7%) |
| | Osaka | 66 (6.6%) | 37 (3.7%) | 29 (1.7%) |
| | Fukuoka | 103 (10.3%) | 39 (3.9%) | 64 (6.4%) |

versions: SOGS (Lifetime), PGSI, LieBet, and DSM-5 (12 months). Additionally, the Japanese government's survey on gambling conditions used the Lifetime time frame for the SOGS, which was adopted for this study.

A chi-square test was conducted to determine the differences in prevalence at the original cut-off scores of the four screening tests by sex and region; the cut-off scores were as follows: scores ≥ 5 points for the SOGS, scores ≥ 8 points for the PGSI, scores ≥ 1 point for the LieBet, and scores ≥ 4 points for the DSM-5. In this study, we define a status that exceeds the cut-off scores as "possible gambling disorder". The distribution of prevalence for these four screening tests was then graphically illustrated in a 'dango' chart. The statistical analyses were performed using SPSS version 27.

Permission to conduct this study was sought from the Ethics Review Board of Nara Medical University, which approved the study (Approval No: 2892). Participation in this study was voluntary, and all respondents provided electronic consent. The procedures used in this study adhere to the tenets of the Declaration of Helsinki.

## Results

Table 1 shows respondents' distribution by age group and proportion of male respondents. The division of our sample roughly matched the age and sex percentages in Osaka and Fukuoka prefectures at the time of the survey.

Table 2 shows the number and percentage of participants who engaged in the eight gambling activities in Japan. The gambling activities are listed in order of total participation, from high to low. The gambling activity with the largest number of participants was the lottery, in which more than half of the respondents had participated. Approximately 10% of the respondents had frequented casinos outside Japan.

Table 3 shows the prevalence of possible gambling disorders based on the original cut-off scores of the four screening tests by total sample, sex, and region. Regarding the total sample, the prevalence was higher for the SOGS (n = 205, 10.3%), followed by the PGSI (n = 134, 6.7%), LieBet (n = 97, 4.9%), and DSM-5 (n = 75, 3.8%). Significant differences existed between the SOGS and each of the other screening tests. There were also significant differences in the prevalence between PGSI vs. LieBet and PGSI vs. DSM-5. However, it should be noted that the SOGS calculates prevalence over one's lifetime.

The total number of male respondents was significantly higher than that of female respondents for all four screening tests. The prevalence of possible gambling disorders among male respondents was approximately four times higher than that of female respondents on all four tests. Regarding the total number of people with a possible gambling disorder by region, significant differences were found for the SOGS and DSM-5, with the total number of respondents in Osaka being significantly higher than that in Fukuoka.

Fig 1 graphically shows the prevalence of four screening tests. We call this chart a 'dango' chart because its shape is similar to that of a *dango*, a traditional Japanese rice sweet. As aforementioned, there is a nearly threefold difference in the prevalence at the original cut-off scores between DSM-5 (3.8%) and the SOGS (10.3%), and a nearly twofold difference between DSM-5 and the PGSI (6.7%). In addition, while 70.2% of the respondents had a SOGS score of 0, this percentage was largely different from that for the LieBet (95.1%). For the SOGS, changing the cut-off from 5 to 4 increases the prevalence by 2.9%, while changing the cut-off

**Table 3. Number and percentage above the original cut-off score for four different screening tests by the total sample, sex, and region.**

|  | SOGS (score ≥ 5) | | PGSI (score ≥ 8) | | LieBet (score ≥ 1) | | DSM (score ≥ 4) | |
|---|---|---|---|---|---|---|---|---|
|  | Lifetime | | 12 months | | 12 months | | 12 months | |
| **Total**[a] | 205 | (10.3%) | 134 | (6.7%) | 97 | (4.9%) | 75 | (3.8%) |
| **Male** | 164** | (16.8%) | 105** | (10.7%) | 78** | (8.0%) | 61** | (6.2%) |
| **Female** | 41 | (4.0%) | 29 | (2.8%) | 19 | (1.9%) | 14 | (1.4%) |
| **Male/Female** | 4.2 | | 3.8 | | 4.2 | | 4.4 | |
| **Osaka** | 119 | (11.9%) | 73 | (7.3%) | 56 | (5.6%) | 47 | (4.7%) |
| **Fukuoka** | 86* | (8.6%) | 61 | (6.1%) | 41* | (4.1%) | 28* | (2.8%) |

[a]All combinations except the one between LieBet and DSM-5 were significant.

** p < .001, * p < .05

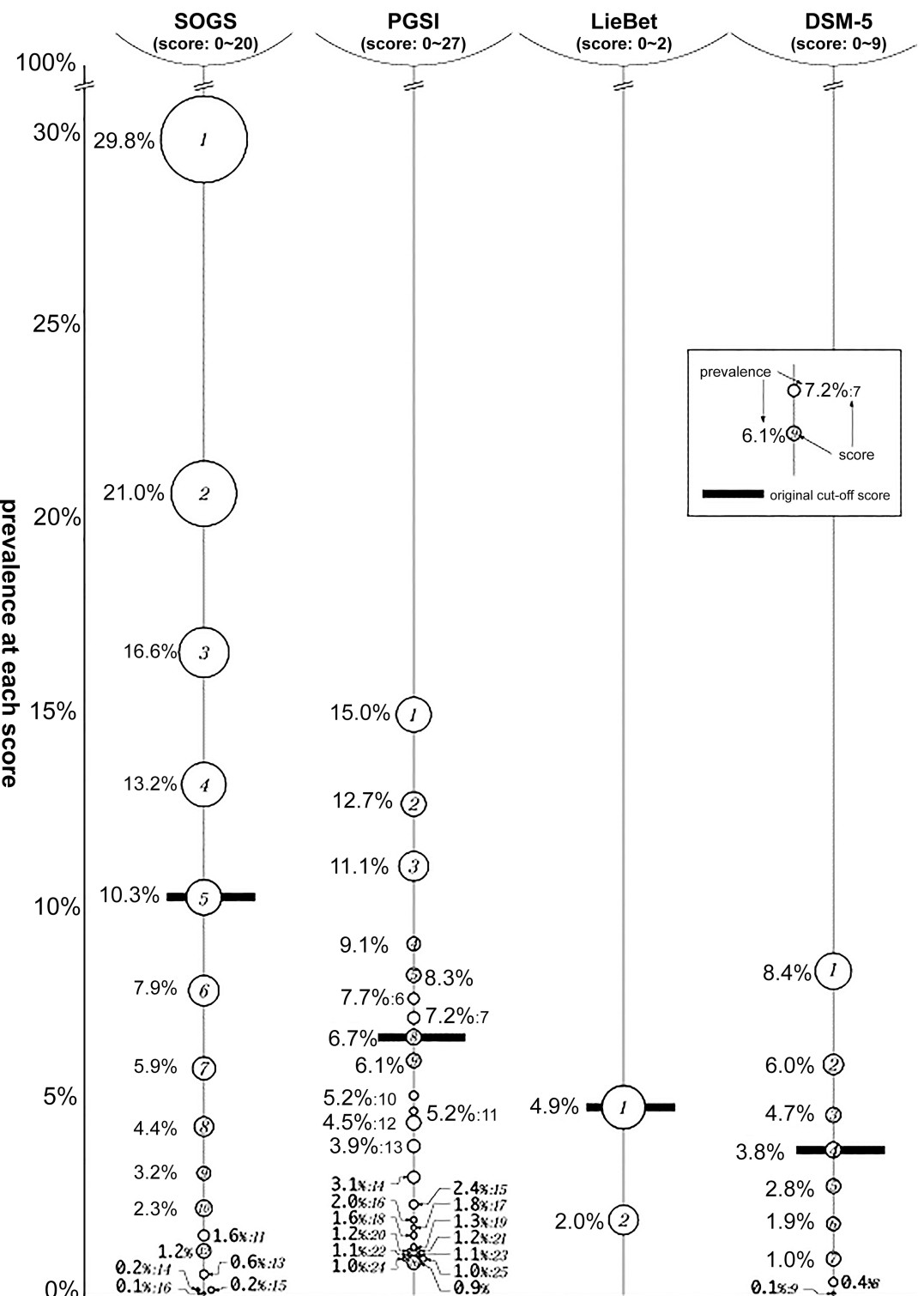

**Fig 1. Prevalence across the four screening tests (dango chart).** Numbers in circles: scores. Percentages near circles: prevalence at each score. Size of circles: proportional to the number of people in each score. Arc at the top: proportional to the number of people who scored zero. Bold line: original cut-off scores for each test.

from 8 to 7 for the PGSI increases the prevalence by only 0.5%, and changing the cut-off to 4 increases it by 2.4%.

## Discussion

The selection of the screening tests, cut-off scores, and survey mode are important topics in surveys on the prevalence of possible gambling disorders, yet few researchers have focused on this issue, and survey design and implementation have received little attention in this regard [29]. We examined the impact of the choice of screening method and cut-off score on the prevalence of possible gambling disorder by administering four screening tests to the same respondents at the same time.

First, the SOGS showed the highest prevalence at the original cut-off scores, followed by the PGSI, LieBet, and DSM-5. Regarding the selection of screening tests, the SOGS screened out more individuals with a possible gambling disorder than the other measures. As for differences by sex, all tests showed that men are approximately four times more likely to suffer from a possible gambling disorder than women; this finding is similar to the results of previous studies [30,31]. Given that the differences by sex were similar across all four tests, our results suggest that the screening test had little influence on the differences in possible gambling disorder prevalence by sex. Finally, the prevalence between the two regions of Osaka and Fukuoka was significantly different for the SOGS and DSM-5, but not for the PGSI and LieBet.

Second, when comparing the distribution of prevalence of possible gambling disorder by the different cut-off scores, we found that for the SOGS, changing the cut-off score from 5 to 4 increased the prevalence by 2.9%, while changing that for the PGSI from 8 to 7 increased the prevalence by 0.5%. For the DSM-5, changing the cut-off score from 4 to 3 increased the prevalence by only 0.9%. Hence, changing the cut-off score had a large impact on the prevalence of the SOGS but only a small impact on the prevalence of the PGSI and DSM-5. These results are consistent with those of prior research, which reports more false positives for the SOGS than for other screening tests [29].

In this study, we proposed a new chart (the dango chart) as a method for presenting results when several screening tests are administered to the same sample. This chart provides a visual explanation of the characteristics of each screening test based on the scatter pattern (score range and interval) of the score circles. The range of the prevalence of respondents scoring 1 or higher is wide for the SOGS (0.1–29.8%), but narrower for the PGSI (0.9–15.0%) and DSM-5 (0.1–8.4%), and clustered on the severe side, indicating that the SOGS comprises questions that the general population is more likely to answer than the other two. The dango chart shown in this paper is useful for comparing several screening tests because the chart makes it easy to understand whether a certain score on a given test is equivalent to that on another test.

In this study, the prevalence for the PGSI was 0.6 times lower than that for the SOGS (lifetime) at the original cut-off score. In studies conducted in Canada and Australia with the same subjects, the prevalence of the PGSI was approximately 0.6 times higher than that for the SOGS (one year). Considering the lifetime prevalence of the SOGS in this study, the results of these studies are similar to our findings [32,33].

Although respondents scoring below the cut-off in each test have not received substantial attention, the continuous score distribution in the Dango Chart offers insights into their actual situations. It is plausible that this group includes individuals who may develop gambling-related harm in the future. Regarding alcohol use disorders, the screening, brief intervention, and referral to treatment (SBIRT) approach is widely used as a preventive and comprehensive public health strategy. Adapting SBIRT's concepts and methodologies to

gambling disorders could be beneficial [34]. In alcohol use disorders, appropriate screening enables suitable interventions; a similar prevention-focused approach would be valuable for gambling disorders [35]. To manage the spectrum of gambling-related harm to pathological gambling, long-term monitoring and continuous evaluation methods, such as the Dango Chart, would be advantageous.

The present study has several limitations. First, the responses to the SOGS cover lifetime participation. The prevalence for this test at the original cut-off score also appears to be higher than that for other tests, which in turn cover a one-year period. Because the SOGS originally assessed lifetime prevalence, we chose to use this specific assessment tool in this study. In the future, using the SOGS that covers a one-year period would be preferable [9]. The second limitation is the possible sampling bias inherent to online surveys. The study population could be biased compared with a sample derived from random sampling because online survey respondents are affected by the Internet use environment and incentives and rewards provided for participation [36]. Although this bias did not affect the inter-test comparisons here because of the administration of several tests to the same subjects, its influence on the analysis of cut-off scores (the distribution in the dango chart) is undeniable. The third limitation is the influence of using online surveys as the survey mode. Online surveys tend to produce a higher prevalence of gambling disorders than other survey modes. There has been some argument that non-face-to-face survey modes are more likely to yield honest responses to sensitive questions, such as those on gambling behaviors [37]. Still, even if there was overestimation bias, we believe it did not have a significant influence on the analysis of the inter-test comparisons or cut-off scores because several screening tests were simultaneously administered to the same respondents in this study. The fourth limitation is that the order of the four screening tests was not randomized. The order of administration may have influenced participant responses.

Despite these limitations, this is the first study conducted in Japan to compare the results of screening tests for possible gambling disorders and examine cut-off scores after administering several tests, and its findings demonstrate that the selection of screening tests and cut-off scores makes a significant difference when measuring the prevalence of possible gambling disorders.

In conclusion, the selection of screening tests significantly impacts the prevalence of possible gambling disorders. Although the PGSI now tends to be used more frequently than the SOGS, studies focused on the interchangeability of test results are limited [5]. Hence, the results of different screening tests conducted independently should be carefully compared by researchers. To improve the generalizability of the findings of future studies, scholars should more frequently simultaneously apply several screening tests to the same sample when surveying the prevalence of possible gambling disorders. Further studies using various screening tests simultaneously are recommended to compare the properties of each screening tests.

## Acknowledgments

We appreciate the statistical advice given by Dr. Hidefumi Hitokoto and the graphic design assistance provided by Mr. Kouji Watanabe. We would like to thank Editage (www.editage.com) for English-language editing.

## Author contributions

**Conceptualization:** Tatsuya Noda, Moritoshi Kido, Chieko Ito.

**Data curation:** Tatsuya Noda, Moritoshi Kido.

**Formal analysis:** Tatsuya Noda, Moritoshi Kido, Chieko Ito.

**Supervision:** Toshiyuki Ojima.

**Writing – original draft:** Tatsuya Noda, Moritoshi Kido.

**Writing – review & editing:** Tatsuya Noda, Toshiyuki Ojima.

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
