## [Decision Letter · Decision Letter 0]

23 Jul 2024

PONE-D-24-05925The effect of the choice of screening test when measuring the prevalence of gambling disorder: A cross-sectional study in JapanPLOS ONE

Dear Dr. Noda,

Thank you for submitting your manuscript to PLOS ONE. After careful consideration, we feel that it has merit but does not fully meet PLOS ONE’s publication criteria as it currently stands. Therefore, we invite you to submit a revised version of the manuscript that addresses the points raised during the review process.

We look forward to receiving your revised manuscript.

Kind regards,

José C. Perales

Academic Editor

PLOS ONE

2. In the online submission form, you indicated that [The datasets acquired and analyzed in this study are available upon reasonable request.

All data files are owned by the KAKENHI Research Team on Addiction under the approval of the Ethics Committee of Nara Medical University. Data requests should be sent to the Nara Medical University Research Staff (t-n@umin.ac.jp).].

Reviewers' comments:

Reviewer's Responses to Questions

**Comments to the Author**

1. Is the manuscript technically sound, and do the data support the conclusions?

Reviewer #1: Partly

Reviewer #2: Partly

2. Has the statistical analysis been performed appropriately and rigorously? 

Reviewer #1: Yes

Reviewer #2: N/A

3. Have the authors made all data underlying the findings in their manuscript fully available?

Reviewer #1: Yes

Reviewer #2: Yes

4. Is the manuscript presented in an intelligible fashion and written in standard English?

Reviewer #1: Yes

Reviewer #2: Yes

5. Review Comments to the Author

Reviewer #1: Summary

There is a lot of interesting evidence here, and the dango chart seems useful for comparing psychometric tools. However, I think the narrative of this paper could be developed more. I find myself asking what the main message is and how it informs the field at large. What stands out to me looking at this data is how limited the functionality of gambling screening tools is in that they use largely arbitrary cut-offs to identify risk, as exemplified by the sometimes-drastic differences across the present data. I think there is a reasonable argument to be made that they do not identify the totality of harm and instead focus on those regarded as high risk. From my perspective, the findings tell a meaningful story, but I am not sure that story is discussed as well as it could be. In some areas the narrative and supporting evidence feels rather old and outdated. I have included some papers in my feedback that the authors might find useful.

Broadly speaking, I believe (and the evidence suggests) that gambling screening tools are useful to a point but are inherently flawed as they are derived from a clinical perspective of harm. This leads to them only measuring those with significant issues (even tools like the PGSI). This leads to low prevalence rates, giving the impression of a niche issue. However, harm exists on a continuum and those who sit lower down (low risk, medium risk) are often not accounted for or missed by such tools as they measure expressions of harm and not things predictive of risk. As such, I believe any discussion of the gambling prevalence data needs to have this limitation in mind and should be discussed as part of the work's narrative. If the long-term aim is to track the changes in harm in these two areas as a result of the casino, then the simple application of the SOGS, PGSI (or others CPGI, GABs, NODS, etc.) is not enough to track the changes in risk on a societal level, and a more holistic approach that takes into account risk factors is needed. This point should be made clear in the paper.

I recognise that the authors may not wish to make such an amendment, but it could lead to a paper that meaningfully contributes to the literature.

Line(s) 32: Its not clear why the lie bet was used. it's accuracy and reliability are fairly underwhelming. its only befit is its lengths but there are other tools (like the brief biosocial gambling screen) which are short and more psychometrically robust.

Line(s) 52- 54: Is this context needed here? Feels out of place.

Line(s) 57- 58 & 61 - 63: This source is 12 years old. Given that you are citing prevalence data it should be the most up to date statistics.

Line(s) 78 – 80: I'm not sure I agree. Yes, this would lead to a more complete summary of the prevalence but what would that actually tell you? The key issues with population survey are that they do not help inform harm prevention because the majority of the cases (low risk gamblers) are not identified in the data. In other words, the prevalence only represents the tip of the iceberg. Some commentary on this limitation would be useful.

Line(s) 83: It's also worth noting that gambling screening tools (even the SOGS) are not measuring gambling disorder, they are simply counting symptoms presumed to be indicative of it. As such its more appropriate to discuss this by using the term gambling related harm. Unless there is a diagnosis form a clinician guided by the DSM (or ICD) then the term gambling related harm is more fitting.

Line(s)86: Try to draw on a consistent terminology. Gambling addiction is not listed in the ICD or the DSM. Again, the term gambling related harm is a better umbrella term.

Line(s) 96: Why was Fukuoka selected as a point of comparison? Some justification (even something minor) would be useful.

Line(s) 102 – 103: The PGSI and the Lie bet are not measuring gambling disorder. Amend to gambling related harm.

Line(s) 131: Gambling related harm

Line(s) 137: It's common practice to amend this to 12 month or 6 months. Was there a specific reason why this was not done? You mention in the discussion that this is a limitation but some discussion in the method would be useful as well.

Line(s) 152: This is new to me and is a rather elegant way of presenting this kind of data. However, the figure is low resolution making it harder to read the smaller scores.

Line(s) 167-170: There has recently been a few systematic reviews that have examined this very topic, these should be examined and discussed:

https://doi.org/10.1007/s10389-021-01678-9

https://doi.org/10.1186/s13722-021-00243-9

https://doi.org/10.1016/j.jclinepi.2019.12.022

https://doi.org/10.1016/j.cpr.2019.101784

Line(s) 177: amend to “those experiencing gambling related harm”

Line(s) 188- 198: This is rather old, see below:

https://doi.org/10.1007/s10389-021-01678-9

https://doi.org/10.1177/1073191113500522

Reviewer #2: This study aims to investigate the relationship between scores on two gambling problem screening instruments and symptoms of gambling addiction. While efforts to improve the sensitivity of these instruments are commendable, this study would be strengthened by a more robust theoretical foundation and clearer methodological explanations.

Although the SOGS and PGSI has been traditionally used for screening, are increasingly employed to assess the severity of problems beyond just addiction symptoms. The authors should elaborate on this growing application and clearly define the types of behaviors, problems or symptoms assessed in each instrument, including DSM-5 criteria.

Crucially, this study may be limited by a potential order effect, where the order in which the instruments were administered may influence how the data is interpreted. The limited description of the assessment procedure, lacking key details, makes it impossible to determine whether an order effect might have influenced the results. Administering three instruments with very similar questions, without counterbalancing the order, could bias participants' responses. This ambiguity regarding the order effect raises concerns about the trustworthiness of the study's findings.

The study design seems to have originally aimed at tracking gambling prevalence in various Japanese prefectures before and after casinos opened. However, the link between this objective and the chosen measures and test analyses remains unclear. To improve reader understanding, the authors should explain this connection more clearly and provide additional context about gambling practices in Japan.

Minor: A section on measures should be included, explaining all measures and their psychometric properties. Additionally, the section describing the sample and its recruitment procedure should be expanded.

If one of the study's aims is to measure prevalence, then why not use a representative sample? Otherwise, I'm not convinced that the study can be used to make arguments about the effect of gambling availability.

6. PLOS authors have the option to publish the peer review history of their article (what does this mean? ). If published, this will include your full peer review and any attached files.

**Do you want your identity to be public for this peer review?** For information about this choice, including consent withdrawal, please see our Privacy Policy .

Reviewer #1: No

Reviewer #2: No

---

## [Author Response · Author response to Decision Letter 1]

21 Nov 2024

Please refer to “Response to Reviewers” file.

---

## [Decision Letter · Decision Letter 1]

24 Jan 2025

The effect of the choice of screening test when measuring the prevalence of gambling disorder: A cross-sectional study in Japan

PONE-D-24-05925R1

Dear Dr. Noda,

We’re pleased to inform you that your manuscript has been judged scientifically suitable for publication and will be formally accepted for publication once it meets all outstanding technical requirements.

Kind regards,

José C. Perales

Academic Editor

PLOS ONE

Additional Editor Comments (optional):

Reviewers' comments:

Reviewer's Responses to Questions

**Comments to the Author**

1. If the authors have adequately addressed your comments raised in a previous round of review and you feel that this manuscript is now acceptable for publication, you may indicate that here to bypass the “Comments to the Author” section, enter your conflict of interest statement in the “Confidential to Editor” section, and submit your "Accept" recommendation.

Reviewer #1: All comments have been addressed

Reviewer #2: All comments have been addressed

2. Is the manuscript technically sound, and do the data support the conclusions?

Reviewer #1: Yes

Reviewer #2: Yes

3. Has the statistical analysis been performed appropriately and rigorously? 

Reviewer #1: Yes

Reviewer #2: Yes

4. Have the authors made all data underlying the findings in their manuscript fully available?

Reviewer #1: Yes

Reviewer #2: Yes

5. Is the manuscript presented in an intelligible fashion and written in standard English?

Reviewer #1: Yes

Reviewer #2: Yes

6. Review Comments to the Author

Reviewer #1: (No Response)

Reviewer #2: The authors have satisfactorily addressed all the comments raised in the previous review. The revisions have significantly improved the readability and overall quality of the manuscript. The clarity of the arguments and the presentation of the data have been enhanced, making the paper more accessible to a wider audience.

7. PLOS authors have the option to publish the peer review history of their article (what does this mean? ). If published, this will include your full peer review and any attached files.

**Do you want your identity to be public for this peer review?** For information about this choice, including consent withdrawal, please see our Privacy Policy .

Reviewer #1: No

Reviewer #2: No

---

## [Editor Report · Acceptance letter]

PONE-D-24-05925R1

PLOS ONE

Dear Dr. Noda,

I'm pleased to inform you that your manuscript has been deemed suitable for publication in PLOS ONE. Congratulations! Your manuscript is now being handed over to our production team.

Kind regards,

on behalf of

Dr. José C. Perales

Academic Editor

PLOS ONE